# Expression Analysis of Outer Membrane Protein HPS_06257 in Different Strains of *Glaesserella parasuis* and Its Potential Role in Protective Immune Response against HPS_06257-Expressing Strains via Antibody-Dependent Phagocytosis

**DOI:** 10.3390/vetsci9070342

**Published:** 2022-07-06

**Authors:** Xiaojun Chen, Hanye Shi, Xingyu Cheng, Xiaoxu Wang, Zongjie Li, Donghua Shao, Ke Liu, Jianchao Wei, Beibei Li, Jian Wang, Bin Zhou, Zhiyong Ma, Yafeng Qiu

**Affiliations:** 1Shanghai Veterinary Research Institute, Chinese Academy of Agricultural Sciences, Shanghai 200241, China; chenxiaojun159753@163.com (X.C.); 17816899161@163.com (H.S.); xingyucheng1210@163.com (X.C.); lizongjie@shvri.ac.cn (Z.L.); shaodonghua@shvri.ac.cn (D.S.); liuke@shvri.ac.cn (K.L.); jianchaowei@shvri.ac.cn (J.W.); lbb@shvri.ac.cn (B.L.); 2Shanghai Animal Disease Control Center, Shanghai 201103, China; wangxiaoxu1129@163.com (X.W.); jianwhlj@163.com (J.W.); 3College of Veterinary Medicine, Nanjing Agricultural University, Nanjing 210095, China; zhoubin@njau.edu.cn

**Keywords:** *Glaesserella parasuis*, HPS_06257, immunization, antibody-dependent phagocytosis

## Abstract

**Simple Summary:**

*Glaesserella parasuis*, one of the opportunistic pathogens causing Glässer’s disease in piglets, has become a significant concern for pig farmers. Vaccination has been shown to be effective in preventing *Glaesserella parasuis* infection by inducing the protective immune response. Notably, a humoral immune response plays an important role in protection of *Glaesserella parasuis* infection. The mechanism of protection by antibodies has been shown to be associated with antibody-opsonized phagocytosis, which facilitates uptake of *Glaesserella parasuis* by phagocytes such as macrophages. Outer membrane proteins of *Glaesserella parasuis*, as the promising candidates, are often chosen to develop subunit vaccines. HPS_06257 is one of the outer membrane proteins that has been shown to confer protection against *Glaesserella parasuis* infection. However, little is known about the role of HPS_06257 in the protective immune response. We demonstrate that antibody-dependent phagocytosis is involved in the protective effects of HPS_06257. Our findings extend our understanding of how antibody-dependent phagocytosis may contribute to the immune protection afforded by other outer membrane proteins. Thus, our study provides insight into the protective antigens of *Glaesserella parasuis* and useful information for the development of novel vaccines to prevent *Glaesserella parasuis* infection.

**Abstract:**

HPS_06257 has been identified as an important protective antigen against *Glaesserella parasuis* infection. However, little is known about the role of HPS_06257 in the protective immune response. A whole-genome data analysis showed that among 18 isolates of *Glaesserella parasuis*, 11 were positive for the *HPS_06257* gene, suggesting that not every strain contains this gene. We used PCR to investigate the presence of the *HPS_06257* gene among 13 reference strains and demonstrated that 5 strains contained the gene. A polyclonal antibody against HPS_06257 was generated with a recombinant protein to study the expression of HPS_06257 in those 13 strains. Consistent with the PCR data, five strains expressed HPS_06257, whereas eight strains were HPS_06257 null. We also compared the protective effects of HPS_06257 against an HPS_06257-expressing strain (HPS5) and an HPS_06257-null strain (HPS11). Immunization with HPS_06257 only protected against HPS5 and not HPS11. Moreover, phagocytosis of antibody-opsonized bacteria demonstrates that the antibody against HPS_06257 increased the phagocytosis of the HPS5 strain by macrophages but not the phagocytosis of the HPS11 strain, suggesting that antibody-dependent phagocytosis is responsible for the protective role exerted by HPS_06257 in the immune response to HPS5. Our data also show that the antibody against HPS_06257 increased the phagocytosis of the other HPS_06257-expressing strains by macrophages but not that of HPS_06257-null strains. In summary, our findings demonstrate that antibody-dependent phagocytosis contributes to the protective immune response induced by immunization with HPS_06257 against HPS_06257-expressing strains.

## 1. Introduction

*Glaesserella parasuis* (*G. parasuis*), formerly known as *Haemophilus parasuis* (HPS), is one of the opportunistic pathogens causing Glässer’s disease in piglets, which is characterized by fibrinous polyserositis, polyarthritis, and meningitis [1]. With increasing reports of this disease, *G. parasuis* has become a significant concern for pig farmers. Vaccination has been shown to be effective in preventing *G. parasuis* infection by inducing the protective immune response. Notably, a humoral immune response plays an important role in protection of *G. parasuis* infection. For example, passive immunization of pigs with anti-serum against *G. parasuis* protected against lethal challenge [2]. The mechanism of protection by antibodies has been shown to be associated with antibody-opsonized phagocytosis, which facilitates uptake of *G. parasuis* by phagocytes such as macrophages [3].

Regarding vaccination, as a promising approach to prevention and control of Glässer’s disease, different kinds of vaccines have been developed. While inactive *G. parasuis* vaccines have some limitations, including the different levels of cross-protection between *G. parasuis* serovars, a short time of protection, and so on, they are widely used throughout the world. In order to overcome the limitations of inactivated vaccines, several other methods have been used to develop novel vaccines. Among them, subunit vaccines have been extensively investigated for this purpose [4]. Several different kinds of proteins have been used to develop subunit vaccines [5,6,7,8,9,10]. Notably, outer membrane proteins are often chosen and confer effective protection against *G. parasuis* infection, mainly of homologous strains [5,11,12].

HPS_06257 is one of the outer membrane proteins that has been shown to confer protection against *G. parasuis* infection [5]. It has a molecular weight of 28 kDa and contains a lipoprotein, GNA1870, domain at its C-terminus. Although data have shown that anti-HPS_06257 serum may protect against *G. parasuis* infection, the specific mechanism of this protective effect remains unclear. Antibody-dependent phagocytosis is an important mechanism underlying the protective roles of some immune antigens [9]. However, little is known about the role of antibody-dependent phagocytosis in HPS_06257-mediated protection against *G. parasuis* infection.

Previous *G. parasuis* genome analyses have identified similarities and divergence in the genomes of different *G. parasuis* strains [13,14,15]. Whether HPS_06257 is present in all strains *G. parasuis* remains unknown. In this study, we identified the presence and absence of HPS_06257 in different *G. parasuis* stains and clarified the role of antibody-dependent phagocytosis in the protection it affords against HPS_06257-expressing strains.

## 2. Materials and Methods

### 2.1. Glaesserella parasuis

*Glaesserella parasuis* was recovered from a stock frozen in 20% glycerol and stored at −80 °C, and was grown on tryptic soy agar (TSA; Difco, Detroit, MI, USA) supplemented with 20 μg/mL nicotinamide adenine dinucleotide (NAD; Sigma, St. Louis, MO, USA) and 5% fetal calf serum (FCS; Thermo Fisher Scientific, Shanghai, China) overnight at 37 °C. A colony was picked from the TSA plate and grown in tryptic soy broth supplemented with 20 μg/mL NAD and 5% FCS for about 12 h at 37 °C with shaking.

### 2.2. DNA Isolation and PCR

The 13 *G. parasuis* reference strains for this study were obtained from the China Institute of Veterinary Drug Control [16]. They were representative of 13 different serovars: serovar 1 (as *Glaesserella parasuis* is formerly known as *Haemophilus parasuis*, in this study, *Glaesserella parasuis* serovar 1, simply named HPS1), serovar 2 (HPS2), serovar 3 (HPS3), serovar 4 (HPS4), serovar 5 (HPS5), serovar 6 (HPS6), serovar 8 (HPS8), serovar 9 (HPS9), serovar 10 (HPS10), serovar 11 (HPS11), serovar 13 (HPS13), serovar 14 (HPS14), and serovar 15 (HPS15). All of these strains were grown as described above and harvested for genomic DNA extraction with the TIANamp Bacteria DNA Kit (Tiangen Biotech Co., Ltd., Beijing, China). All 13 serovars were confirmed with a previously reported PCR method [17].

### 2.3. Preparation of Polyclonal Antibody against HPS_06257

Primers (shown in Appendix A) based on the complete genome sequence of *G. parasuis* isolate SH0104 (GenBank accession number NZ_CP024412) were designed to clone the *HPS_06257* gene. The genomic DNA of *G. parasuis* isolate SH0104 was prepared as described above, and PCR was performed to amplify the gene. The amplification product (about 700 bp) was inserted into the *Bam*HI and *Xho*I sites of the pET28a-vector (Novagen, Darmstadt, Germany) to generate the construct pET-HPS_06257. The recombinant HPS_06257 protein (histidine [His]–HPS_06257) was expressed in *Escherichia coli* BL21 cells and purified with the His·Bind Purification Kit (Novagen). A polyclonal antibody against HPS_06257 was then prepared as described previously [18]. Briefly, 8-week-old female BALB/c mice were obtained from JieSiJie Laboratory Animal Co., Ltd. (Shanghai, China) and immunized with His–HPS_06257 combined with complete or incomplete Freund’s adjuvant. After the seventh immunization, sera were collected from the immunized mice. All animal experiments were approved by the Institutional Animal Care and Use Committee of Shanghai Veterinary Research Institute, Chinese Academy of Agricultural Science, Shanghai, China (IACUC no. SHVRI-PO-2016060501) and were performed in compliance with the Guidelines on the Human Treatment of Laboratory Animals (Ministry of Science and Technology of the People’s Republic of China, policy no. 2006398).

### 2.4. Western Blotting

The specificity of the polyclonal antibody against HPS_06257 was examined by Western blotting, as described previously [19]. Briefly, recombinant His–HPS_06257 was transferred to membrane with the control protein, His–SRA (Xiang et al., 2020), and the membrane was blocked with 5% skim milk for 1 h at room temperature. The membrane was incubated overnight at 4 °C with the anti-HPS_06257 antibody (1:1000) or anti-His antibody (1:1000; Sigma), and then incubated for 1 h at room temperature with a horseradish peroxidase (HRP)–conjugated secondary antibody. The membrane was then treated with HRP substrate, according to the instructions for Enhanced Chemiluminescence (ECL) Reagent (Pierce) and exposed to x-ray film. Images were captured with the Gel Doc™ EZ System (Bio-Rad Laboratories, Hercules, CA, USA). A whole-cell lysate of each strain of *G. parasuis* was prepared, and the expression of HPS_06257 in the different strains was determined with the methods described above.

### 2.5. Immunization of Mice and Challenge with G. parasuis

Eight-week-old female BALB/c mice were immunized by subcutaneous injection with His–HPS_06257 combined with complete or incomplete Freund’s adjuvant (His–HPS_06257, 60 μg/mouse). Control mice were injected subcutaneously with phosphate-buffered saline (PBS) or adjuvant alone. Fourteen days after the second immunization, the mice were challenged with 2 × 10^9^ colony-forming units (CFU) of HPS5 or HPS11 by intraperitoneal injection. After the challenge, the mice were monitored twice daily and were euthanized after a bodyweight loss of >10%.

### 2.6. Antibody-Mediated Phagocytosis

*G. parasuis* was grown as described above and inactivated with heat. The heat-inactivated bacteria were surface labeled with fluorescein isothiocyanate (FITC), as described previously [20]. The FITC-labeled bacteria (10^8^ cells) were opsonized with 10 μL of serum for 45 min at 37 °C, with occasional agitation. The opsonized bacteria were incubated at a multiplicity of infection of 300 bacteria per RAW264.7 cell for 2 h at 37 °C, as previously described [21]. Trypan blue was then added to quench the extracellular fluorescence. The fluorescence was measured with a microplate fluorometer (at 485 nm excitation/535 nm emission, M3; Molecular Devices, Sunnyvale, CA, USA). Among the PBS-, adjuvant-, and HPS_06257-treated groups of mice, the percentage phagocytosis was calculated by comparing the fluorescence of the adjuvant or HPS_06257 group with the fluorescence of the PBS group (control). Normal mouse serum was also used as a control for the calculation of anti-HPS_06257-antibody-mediated phagocytosis.

### 2.7. Statistical Analysis

All data were analyzed with the GraphPad Prism software version 5.01 (GraphPad Software, Inc., La Jolla, CA, USA). An unpaired Student’s *t* test was used to determine the significance of differences. Data were considered statistically significant at *p* < 0.05. Data are given as means ± standard deviations, as indicated; *n* refers to the sample size.

## 3. Results

### 3.1. Differential Presence of HPS_06257 Gene in Different G. parasuis Strains

The presence of the *HPS_06257* gene was investigated in the whole-genome sequences of *G. parasuis* (Table 1), which have been deposited in GenBank. Among the 18 sequenced strains, 11 strains contained the *HPS_06257* gene sequence, whereas 7 strains lacked the *HPS_06257* gene sequence. In the meanwhile, PalA, Omp2, and D15, the reported antigens [5] of *G. parasuis,* were also analyzed and their gene information is included in Table 1.

To confirm the bioinformatics data, we examined the presence of the *HPS_06257* gene in different *G. parasuis* strains (a total of 13 standard strains) with a PCR assay. Five of these strains (HPS5, HPS10, HPS13, HPS14, and HPS15) contained the *HPS_06257* gene and eight strains (HPS1, HPS2, HPS3, HPS4, HPS6, HPS8, HPS9, and HPS11) did not (Figure 1A–C). The *HPS_06257* gene was confirmed in strain SH0104 (included in Table 1) with gene cloning and plasmid construction. Collectively, our data show that the *HPS_06257* gene occurs in only some strains of *G. parasuis*.

### 3.2. Differential Expression of HPS_06257 Protein in Different Strains of G. parasuis

Although we showed that the *HPS_06257* gene occurs in only some strains of *G. parasuis*, it was unclear whether it was expressed in those strains. Therefore, we developed a polyclonal antibody against the recombinant HPS_06257 protein to detect the expression of native HPS_06257 in those strains. Purified recombinant HPS_06257 expressed in *E. coli* (Figure 2A) was used to immunize mice and generate a polyclonal antibody. To examine the specificity of this antibody, we first investigated its reaction with recombinant HPS_06257 and reported His-SRA [18] with Western blotting. The polyclonal antibody specifically reacted with recombinant HPS_06257 but not with recombinant SRA (Figure 2B). In contrast, an anti-His antibody detected both of these recombinant (His-conjugated) proteins. Thus, we confirmed the successful generation of a polyclonal antibody against the HPS_06257 protein.

The polyclonal antibody was then used to determine the expression of the HPS_06257 protein in different strains of *G. parasuis*. Our results showed that HPS_06257 was only expressed by strains containing the *HPS_06257* gene (Figure 2C), consistent with the PCR results. The protein expressed by each strain was detected with SDS-PAGE (Figure 2D). These data confirm the existence of HPS_06257-expressing *G. parasuis* strains and HPS_06257-null *G. parasuis* strains in the field.

### 3.3. Differential Immunoprotective Effects of HPS_06257 against HPS_06257-Expressing Strains and HPS_06257-Null Strains

Because HPS_06257 is an important protective antigen against *G. parasuis* infection, we investigated the immunoprotective effects of HPS_06257 against HPS_06257-expressing strains and HPS_06257-null strains. We examined the immunoprotective role of HPS_06257 against HPS5 (an HPS_06257-expressing strain) and HPS11 (an HPS_06257-null strain) infections. Fourteen days after the booster immunization, each group was challenged with five median lethal doses of HPS5 or HPS11. The mice immunized with HPS_06257 showed 80% survival within 72 h of challenge with HPS5, whereas the mice treated with PBS or adjuvant all died within 60 h of challenge with HPS5 (Figure 3A). All the mice challenged with HPS11 died within 36 h of challenge (Figure 3B). These results indicate that HPS_06257 is protective against infection with HPS5 (an HPS_06257-expressing strain) and not against HPS11 (an HPS_06257-null strain).

### 3.4. Antibody-Dependent Phagocytosis Is Associated with the Immunoprotective Effects of HPS_06257 against HPS_06257-Expressing Strains

Antibody-dependent phagocytosis has been shown to contribute to the protective effects of antigens against microbial infections. However, whether antibody-dependent phagocytosis is associated with the immunoprotective effects of HPS_06257 against HPS_06257-expressing strains was unclear. Therefore, we next examined the effect of the mouse serum isolated from each group in the challenge experiment on the phagocytosis of *G. parasuis* by macrophages. The sera from HPS_06257-immunized mice increased the phagocytosis of HPS5, whereas the sera from mice not immunized with HPS_06257 (PBS or adjuvant group) did not, consistent with the protective effect of HPS_06257 against HPS5 challenge (Figure 4). In contrast, the sera from HPS_06257-immunized mice did not affect the phagocytosis of HPS11 compared with the serum from unimmunized mice (PBS or adjuvant group) (Figure 4). These results indicate that the sera from HPS_06257-immunized mice increased the phagocytosis of HPS5 but not HPS11, consistent with the protective effect of HPS_06257 against HPS_06257-expressing strains but not against HPS_06257-null strains.

To confirm the effect of the anti-HPS_06257 antiserum on phagocytosis by macrophages, we examined effects of the polyclonal antibody against HPS_06257 on the 13 different strains shown in Figure 1C compared with the effects of normal mouse serum treated bacteria. The anti-HPS_06257 serum significantly increased the phagocytosis of all the HPS_06257-expressing strains compared with the treatment of some of these strains with normal mouse serum (Figure 5). However, the anti-HPS_06257 serum had no effect on the phagocytosis of any HPS_06257-null strain compared with the effect normal mouse serum had on some of these strains (Figure 5). Collectively, these findings demonstrate that antibody-dependent phagocytosis contributes to the immunoprotective effect of HPS_06257 against infections with HPS_06257-expressing *G. parasuis* strains.

## 4. Discussion

This study provides novel information on the expression characteristics of HPS_06257 and its role in the development of a protective immune response against *G. parasuis* infection. Using a genomic analysis, PCR, and Western blotting, we demonstrated the presence of HPS_06257-expressing *G. parasuis* strains and HPS_06257-null *G. parasuis* strains. Furthermore, by comparing the immune protection afforded by HPS_06257 against HPS_06257-expressing and HPS_06257 null strain infections, we showed that antibody-dependent phagocytosis contributes to the protective immune response induced by HPS_06257 immunization against HPS_06257-expressing strains. Collectively, these data provide novel evidence for the role of HPS_06257, an important immune antigen, in the protective immune response to *G. parasuis* infection.

Since HPS_06257 was identified in an immunoproteomic analysis [3], it has been studied as an immune antigen effective against the parental *G. parasuis* infection. Regarding the strong identity shared by the protein sequences of different strains (based on the BLASTP analysis, data not shown), HPS_06257 would be a good candidate for a protective antigen for subunit vaccine development. In this study, we demonstrated the existence of HPS_06257-expressing and HPS_06257-null *G. parasuis* strains. Thus, this finding gives some clue that HPS_06257 could not protect against all of strains in the field despite of the strong identity shared by the HPS_06257 protein sequences of different strains.

A previous study showed that antiserum against HPS_06257 had bactericidal activity in a whole-blood killing assay [5]; however, the specific mechanism of this protective effect remains unclear. In the present study, we demonstrated that antibody-dependent phagocytosis contributes to the protective effect induced by HPS_06257, providing further insight into the protective effect mediated by anti-HPS_06257 antiserum.

Because HPS_06257 was identified as an outer membrane protein in an immunoproteomic analysis, it is not unexpected that antibody-dependent phagocytosis is involved in its involvement in the protective immune response it induces against *G. parasuis* infection. The immunogenic actions of bacterial outer membrane proteins have been extensively studied [11,12,22]. Notably, antibody-dependent phagocytosis has been investigated as the underlying mechanism of the protective immune response [23]. A study of the transferrin binding protein B (TbpB) expressed on the surface of *G. parasuis* showed that antibody-dependent phagocytosis contributes to its protective role in the development of immune response against *G. parasuis* infection [10]. Therefore, antibody-dependent phagocytosis could be an important mechanism underlying the role of HPS_06257 in the protective humoral immune response to *G. parasuis* infection.

In summary, this is the first study to demonstrate that antibody-dependent phagocytosis is involved in the protective effects of HPS_06257. These findings extend our understanding of how antibody-dependent phagocytosis may contribute to the immune protection afforded by other outer membrane proteins. Thus, future studies are needed to further verify this. In the meanwhile, these findings also provide insight into the protective antigens of *G. parasuis* and useful information for the development of a subunit vaccine to prevent *G. parasuis* infection.

## Figures and Tables

**Figure 1 vetsci-09-00342-f001:**
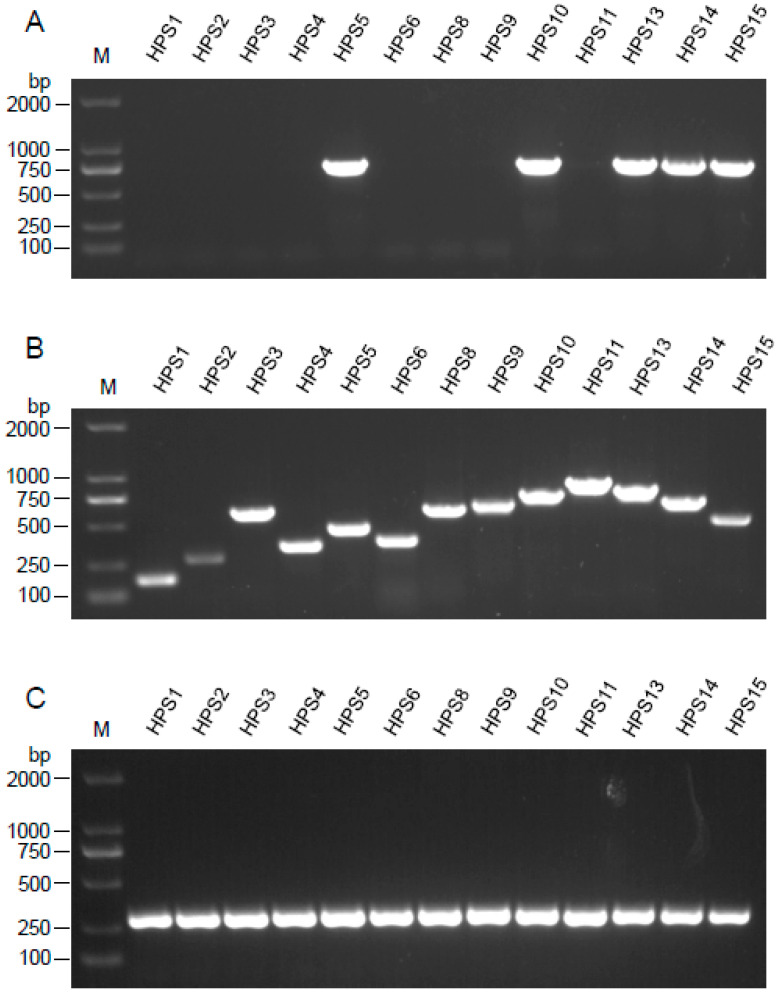
**Detection of *HPS_06257* gene in different *G. parasuis* strains.** Genomic DNA was extracted from different strains of *G. parasuis* for PCR analysis. (**A**) *HPS_06257* gene in different strains of *G. parasuis* was detected with PCR. (**B**) PCR assay was used to identify different serotypes of *G. parasuis*. (**C**) The *HPS_219690793* housekeeping gene was detected with PCR in all the strains described as above.

**Figure 2 vetsci-09-00342-f002:**
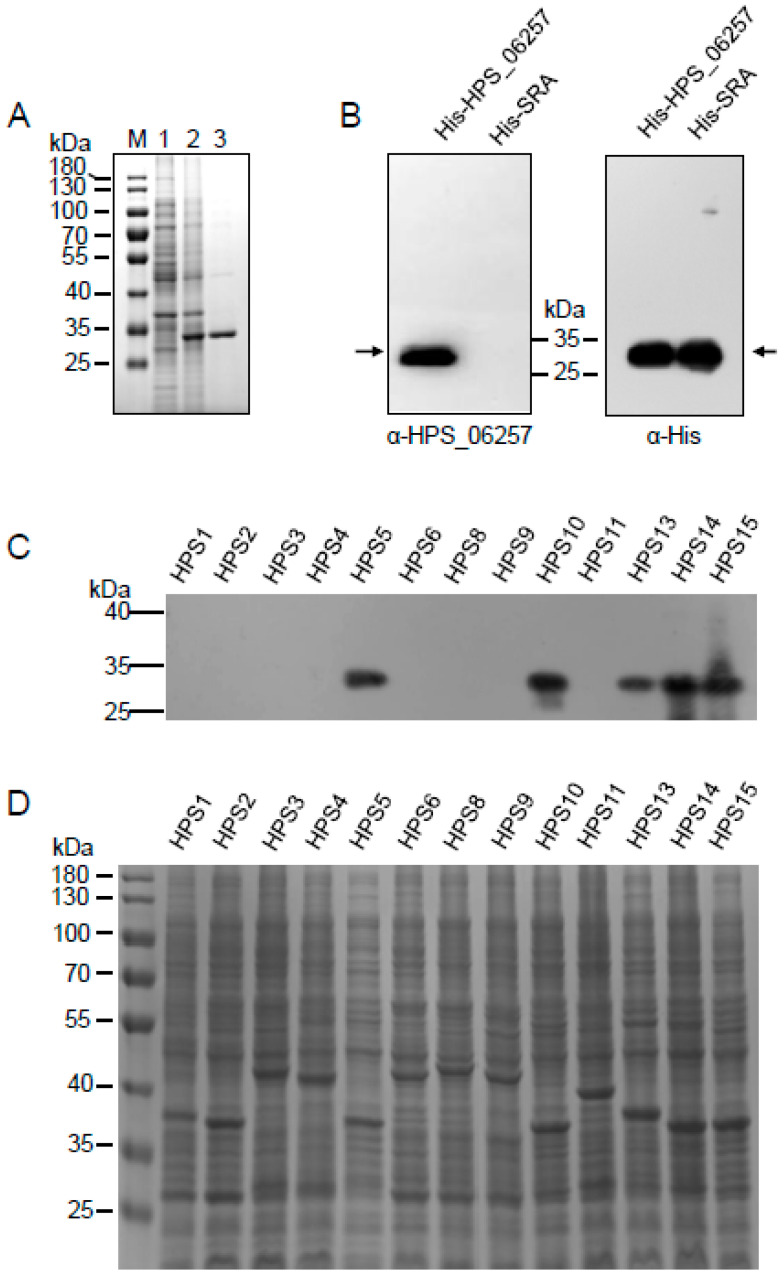
**Detection of HPS_06257 protein expression in different strains of *G. parasuis*.** (**A**) SDS-PAGE gel shows the expression of HPS_06257 protein in *E. coli* (lane 2) and purified HPS_06257 (lane 3). Lane M, protein molecular weight marker; lane 1, *E. coli* transformed with empty control vector. (**B**) Polyclonal antibody against HPS_06257 was generated by immunizing BALB/C mice with purified HPS_0625. Western blotting was performed to determine the specificity of the anti-HPS_0625 antibody. The anti-HPS_0625 antibody specifically recognized recombinant His–HPS_0625 protein but not His–SRA; in contrast, an anti-His antibody specifically recognized both recombinant proteins. Black arrows indicate the target bands. (**C**) Expression of HPS_0625 in different strains of *G. parasuis* was determined with Western blotting using anti-HPS_0625 antibody. (**D**) SDS-PAGE gel shows the total cell lysates of different strains of *G. parasuis*, which were used for the Western blotting analysis.

**Figure 3 vetsci-09-00342-f003:**
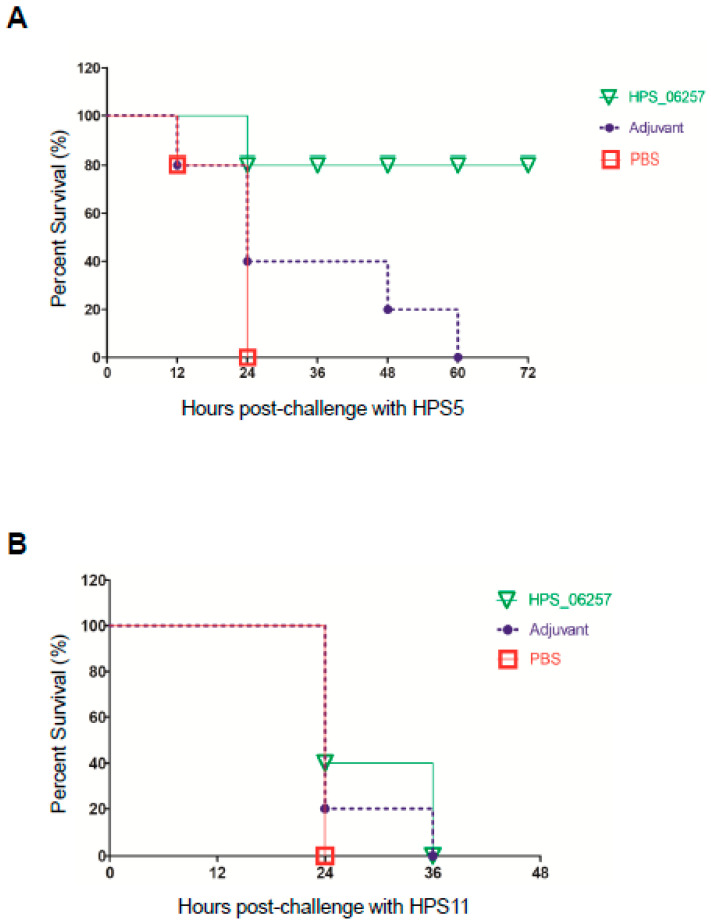
**Comparative analysis of immunoprotective effects of HPS_06257 against *G. parasuis* strains HPS5 and HPS11.** (**A**) Immunoprotective effect of HPS_06257 against HPS5 infection. Mice were immunized with PBS, adjuvant alone, or HPS_06257. The different groups of mice were challenged by intraperitoneal inoculation with 2 × 10^9^ CFU of HPS5. Notably, the HPS_06257 group showed 80% survival after challenge; in contrast, the mice in neither the PBS nor adjuvant-alone group survived after challenge. (**B**) Immunoprotective effects of HPS_06257 against HPS11 infection. Mice were immunized as described above. The different groups of mice were challenged by intraperitoneal inoculation with 2 × 10^9^ CFU of HPS11. Notably, none of the groups showed a protective effect against HPS11 infection.

**Figure 4 vetsci-09-00342-f004:**
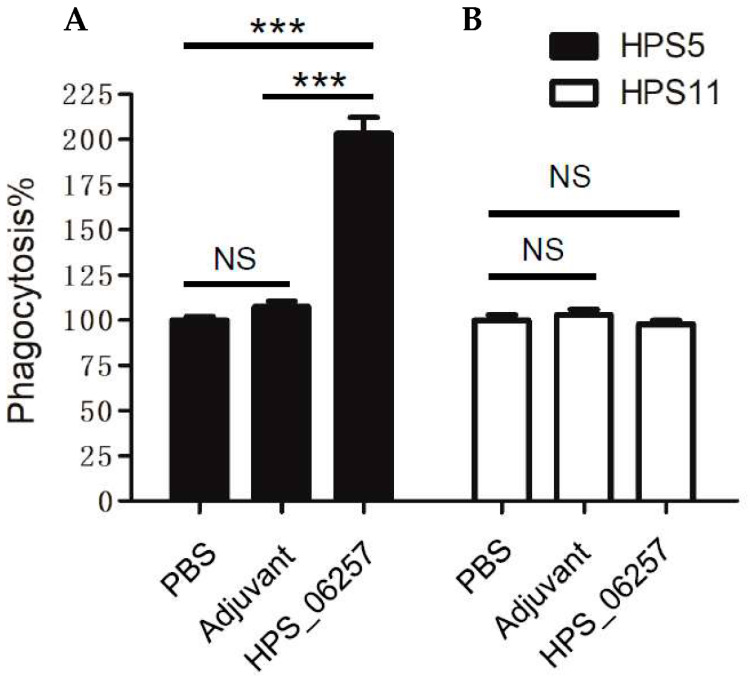
**Comparative analysis of effects of different mouse antisera on phagocytosis of *G. parasuis* strains HPS5 and HPS11.** (**A**) HPS5 was opsonized with mouse antisera from the PBS, adjuvant-alone, or HPS_06257-treated group, and phagocytosis was analyzed in RAW264.7 cells, as described in the Methods section. Notably, the antiserum from the HPS_06257 group significantly promoted the phagocytosis of HPS5 by macrophages, whereas the antiserum from neither the PBS group nor adjuvant group did so. (**B**) HPS11 was opsonized with mouse antisera from the different groups, as described above, and phagocytosis was analyzed in RAW264.7 cells, as described in the Methods section. Notably, the mouse antiserum from the HPS_06257 group did not enhance the phagocytosis of HPS11 by macrophages compared with the effects of antiserum from either the PBS group or adjuvant group. Data are means ± SD of data pooled from one independent experiment; *n* ≥ 3 for each of the analyzed parameters. NS, not statistically significant. *** *p* < 0.001 in a comparison of the PBS and HPS_06257 groups or of the adjuvant-alone and HPS_06257 groups.

**Figure 5 vetsci-09-00342-f005:**
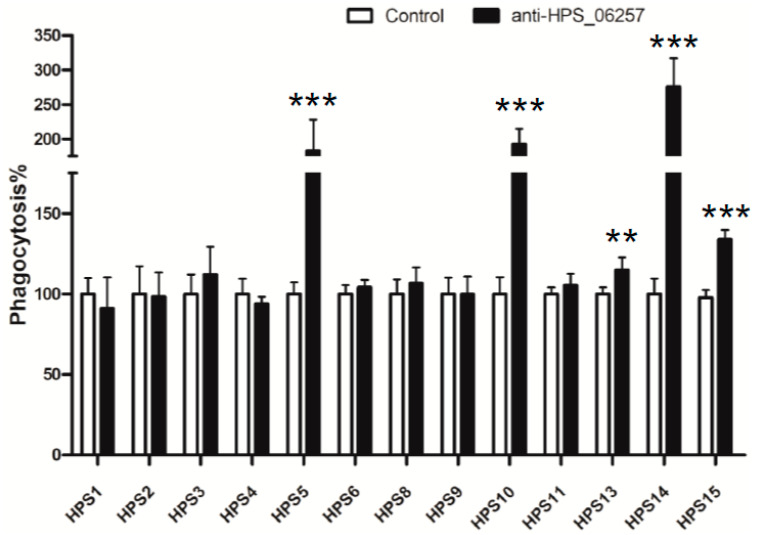
**Effects of anti-HPS_06257 antibody on phagocytosis by macrophages of different strains of *G. parasuis*****.** Different strains of *G. parasuis* were opsonized with anti-HPS_06257 antibody, as shown in Figure 2, or control antibody (normal mouse serum). Notably, compared with control-antibody-opsonized phagocytosis, the anti-HPS_06257 antibody only affected the phagocytosis of HPS_06257-expressing strains. Data are means ± SD of data pooled from two independent experiment; *n* ≥ 6 for each of the analyzed parameters. ** *p* < 0.01; *** *p* < 0.001 in a comparison of the control and anti-HPS_06257 groups.

**Table 1 vetsci-09-00342-t001:** Genome analysis reveals the differential existence of HPS_06257 gene in different strains of *Glaesserella parasuis*.

Strain	GenBank Accession No.	Assembly Status	Serovar	Location	HPS_06257	PalA	Omp2	D15
SH0104	NZ_CP024412	Whole genome	5	China	+	+	+	+
YHP170504	NZ_CP054195	Whole genome	/	China				
HPS412	NZ_CP041334	whole genome	/	China	+	+	+	+
aHPS7	NZ_CP049090	Whole genome	7	China		+	+	+
SCW0912	NZ_CP046114	Whole genome	5	China	+	+	+	+
HPS-1	NZ_CP040243	Whole genome	/	China	+	+	+	+
CL120103	NZ_CP020085	Whole genome	/	China		+		+
KL0318	NZ_CP009237	Whole genome	/	China	+	+	+	+
SC1401	NZ_CP015099	Whole genome	/	China		+		+
sHPS7	NZ_CP049088	Whole genome	7	China		+	+	+
vHPS7	NZ_CP049089	Whole genome	7	China		+		+
GZ20170512	NZ_CP029150	Whole genome	/	China	+	+	+	+
29755	NZ_CP021644	Whole genome	5	USA	+	+	+	+
D74	NZ_CP018032	Whole genome	9	Sweden		+		+
SH0165	NC_CP001321	Whole genome	5	China	+	+	+	+
ZJ0906	NC_CP005384	Whole genome	12	China	+	+	+	+
Nagasaki	NZ_CP018034	Whole genome	5	Japan	+	+	+	+
SH03	NZ_CP009158	Whole genome	/	China	+	+	+	+

## Data Availability

Data will be made available upon request to the authors.

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
