# Peer review of "Expression Analysis of Outer Membrane Protein HPS_06257 in Different Strains of Glaesserella parasuis and Its Potential Role in Protective Immune Response against HPS_06257-Expressing Strains via Antibody-Dependent Phagocytosis"

_vetsci, 2022, doi:10.3390/vetsci9070342_

Round 1

Reviewer 1 Report

In this manuscript, the authors have studied the expression of HPS_06257 in several strains of Glaesserella parasuis. Only 5 serovars, over 13, expressed HPS_06257 and immunization against HPS_06257 protects from servovar expressing HPS_06257. The authors also showed that antibodies against HPS_06257 specifically promote phagocytosis of strains expressing HPS_06257.

The experiments and the results are clearly presented and the authors showed that HPS_06257 is therefore not the best candidate for a vaccine against G. parasuis.

1. In their genomic analysis, the authors could also have studied the presence of others proteins which could be vaccine candidates, such as PalA, Omp2 or D15

2. The authors may provide the sequences of primers used.

3. There is not enough description of Figure 1B. Which gene is used for serotype identification?

Typo error in line 270, "7" is missing in "HPS_0625"

Author Response

In this manuscript, the authors have studied the expression of HPS_06257 in several strains of Glaesserella parasuis. Only 5 serovars, over 13, expressed HPS_06257 and immunization against HPS_06257 protects from servovar expressing HPS_06257. The authors also showed that antibodies against HPS_06257 specifically promote phagocytosis of strains expressing HPS_06257.The experiments and the results are clearly presented and the authors showed that HPS_06257 is therefore not the best candidate for a vaccine against G. parasuis.

  1. In their genomic analysis, the authors could also have studied the presence of others proteins which could be vaccine candidates, such as PalA, Omp2 or D15

Response: We thank the reviewer’s suggestion. We have performed new analysis to determine the presence of other proteins such as PalA, Omp2 and D15. The information about Pal A, Omp2 and D15 has been included into the new Table 1 of the revised manuscript.

  1. The authors may provide the sequences of primers used.

Response: The primers used to amplify HPS_06257 gene has been included in the Table S1. Besides, the PCR primers used to identify molecular serotyping of Glaesserella parasuis have been reported by a paper, which has been cited in the maintext.

  1. There is not enough description of Figure 1B. Which gene is used for serotype identification?

Response: The gene for serotype identification had been added in the revised figure lB legends as “To note that funB, wzx, glyC, wciP, wcwK, gltI, scdA, funV, funX, amtA, gltP, funAB, and funI genes was amplified by PCR as shown in the literature [17].”.

  1. Typo error in line 270, "7" is missing in "HPS_0625"

Response: Yes, you are correct. It has been improved.

Reviewer 2 Report

Chen et al. attempted to evaluate the outer membrane protein HPS_06257 role in the immune response, with a focus on the fact that the antibody against HPS_06257 increased the phagocytosis of other HPS_06257 expressing strains by macrophages, but not that of HPS_06257-null strains. It was stated that the antibody-dependent phagocytosis contributes to the protective immune response induced by immunization with HPS_06257 against HPS_06257-expressing strains...

The title and abstract are representative.

Introduction is too brief, data regarding Glaesserella parasuis interaction with the immune system should be detailed. HPS_06257 structure and functional role should be presented in a clear and simple manner, fully covering its description

2. Materials and Methods

2.1.Glaesserella parasuis

Strain source?

2.3. Preparation of polyclonal antibody against HPS_06257

Eu guidelines differ from Chinese ones regarding animal treatment. The conditions of the mentioned guidelines is therefore unknown for most if not all non-CHN researchers.

The work could have added value to the current literature regarding the topic however insufficient data is presented for such an novo topic.

 I strongly recommend not to use mammals in experiments unless the research work structure is of high complexity and it is absolutely needed. I’m sorry but I consider Major to Reject, there are not enough assays performed to jump to the mentioned conclusions.

Author Response

Chen et al. attempted to evaluate the outer membrane protein HPS_06257 role in the immune response, with a focus on the fact that the antibody against HPS_06257 increased the phagocytosis of other HPS_06257 expressing strains by macrophages, but not that of HPS_06257-null strains. It was stated that the antibody-dependent phagocytosis contributes to the protective immune response induced by immunization with HPS_06257 against HPS_06257-expressing strains...

The title and abstract are representative.

Introduction is too brief, data regarding Glaesserella parasuis interaction with the immune system should be detailed. HPS_06257 structure and functional role should be presented in a clear and simple manner, fully covering its description

Response: We really appreciated the reviewer’s suggestion. We have carefully revised the introduction and added some information of humeral immune response against G. parasuis, as shown in the revised version, “Vaccination has been shown to be effective in preventing G. parasuis infection by inducing the protective immune response. Notable, a humoral immune response plays an important role in protection of G. parasuis infection. For example, passive immunization of pigs with anti-serum against G. parasuis protected against lethal challenge [2]. The mechanism of protection by antibodies has been show to be associated with the antibody-opsonized phagocytosis, which facilitate uptake of G. parasuis by phagocytes like macrophages [3].”

“Regarding on vaccination, as the promising approach to prevention and control of Glässer’s disease, different kinds of vaccines have been developed. While inactive G. parasuis vaccines have some limitation including the different levels of cross-protection between G. parasuis serovars, a short time of protection and so on, they are widely used throughout the world. In order to overcome the limitation of inactivated vaccines, several other methods have been used to develop the novel vaccines. Among them, subunit vaccines have been extensively investigated for this purpose [2].”

In the meanwhile, we tried to provide more information of HPS_06257, unfortunately there are limited information about HPS_06257 structure and functional role except its protective role in induction of immune response against G. parasuis infection.

  1. Materials and Methods

2.1.Glaesserella parasuis

Strain source?

Response: The 13 G. parasuis reference strains for this study were obtained from the China Institute of Veterinary Drug Control, as shown in the revised version.

2.3. Preparation of polyclonal antibody against HPS_06257

Eu guidelines differ from Chinese ones regarding animal treatment. The conditions of the mentioned guidelines is therefore unknown for most if not all non-CHN researchers. The work could have added value to the current literature regarding the topic however insufficient data is presented for such an novo topic.

Response: We understand what the reviewer mentioned. As shown in our manuscript, all animal experiments were approved by the Institutional Animal Care and Use Committee of Shanghai Veterinary Research Institute, Chinese Academy of Agricultural Science, Shanghai, China (IACUC no. SHVRI-PO-2016060501) and were performed in compliance with the Guidelines on the Human Treatment of Laboratory Animals (Ministry of Science and Technology of the People’s Republic of China, policy no. 2006398).

I strongly recommend not to use mammals in experiments unless the research work structure is of high complexity and it is absolutely needed. I’m sorry but I consider Major to Reject, there are not enough assays performed to jump to the mentioned conclusions.

Response: In this study, we used BALB/c or C57BL/6 mice, one of the most popular animal models to generate polyclonal antibodies and perform survival analysis, respectively. Furthermore, we carefully revised our manuscript to refine our conclusion as “In summary, this is the first study to demonstrate that antibody-dependent phagocytosis is involved in the protective effects of HPS_06257. These finding extend our understanding of how antibody-dependent phagocytosis may contribute to the immune protection afforded by other outer membrane proteins. Thus, future studies are needed to further verify this. In the meanwhile, they also provide insight into the protective antigens of G. parasuis and useful information for the development of a subunit vaccine to prevent G. parasuis infection.”.

Reviewer 3 Report

Xiaojun Chen et al., investigated the HPS_06257 protein of Glaesserella parasuis presence and its potential role in protective immune response against HPS5 using antibody-dependent phagocytosis study. Overall, the manuscript was written well and clearly explained all the procedures. Since, the article needs few corrections and clarifications for the improvement.

Major corrections

1. The authors should explain what is HPS_06257-expressing strain (HPS5)/HPS_06257-null (HPS11) and its role in the introduction. Also, the impact of these proteins on the immune response? The authors explained few points in the abstract but not included in the introduction part.

2. Line 42: Missed to inform what is HPS? Haemophilus parasuis serovar? Indicate in the main text.

3. Line 72: The authors were mentioned about the primers, but I could not find in the table S1 file with this article. I found only the main text figures in the supplementary files. The authors must provide the primers details in the supplementary files.

4. Line 132: 11 stains contained the HPS_06257? Table 1 showed only 10 (+) symbols? Please verify it.

5. Line 184, 205, 212, 219 and 229: simplify as HPS5 than HPS_06257-expressing stains the results and discussion part.

6. Figure 4 and 5 not in the chronological order, verify it.

7. Line 268: The authors must provide the Multiple sequence alignment and the percentage of identity in the supplementary section or need citation regarding the sentence "strong identity shared by the HPS_06257".

8. Discussion needs the implementation of the study and benefits of the animals welfare.

Minor corrections

1. Table 1: Write properly "Whole", some in small letters

2. Line 175-179 in page 6 and page 11: Error "HPS_0625" correct and verify it.

Author Response

Xiaojun Chen et al., investigated the HPS_06257 protein of Glaesserella parasuis presence and its potential role in protective immune response against HPS5 using antibody-dependent phagocytosis study. Overall, the manuscript was written well and clearly explained all the procedures. Since, the article needs few corrections and clarifications for the improvement.
Response: We would like to thank Reviewer #3 for their kind words and overall positive response to the manuscript.

Major corrections
1. The authors should explain what is HPS_06257-expressing strain (HPS5)/HPS_06257-null (HPS11) and its role in the introduction. Also, the impact of these proteins on the immune response? The authors explained few points in the abstract but not included in the introduction part.
Response: We really appreciated the reviewer’s suggestion. HPS_06257 has originally been identified by Zhou et al. via an immunoproteome-based approach (Vaccine, 2009, 27(38): 5271-7.). In that study, HPS_06257 as a hypothetical protein was identified and its gene named HPS_06257 gene. Furthermore, based on survival analysis, HPS_06257 was identified as a successful vaccine candidate. We also believe that HPS_06257 is a good vaccine candidate as shown in figure 3A. Interestingly, except on its protectively role in induction of immune response against G. parasuis infection, there is limited information of HPS_06257’s role in G. parasuis. Furthermore, the protective effect of HPS_06257 are less known. Future studies are need to investigate the other characteristic of HPS_06257 instead of a vaccine candidate.

  1. Line 42: Missed to inform what is HPS? Haemophilus parasuisserovar? Indicate in the main text.
    Response: Glaesserella parasuis is formerly known as Haemophilus parasuis. Thus, previous studies are used to call serovar 5 as HPS5 and so on (Vet Microbiol. 2021 Oct;261:109198.; Front Cell Infect Microbiol etc). This is reason that we also the name HPS. In the meanwhile, we have indicated in the revised maintext as “as Glaesserella parasuis is formerly known as Haemophilus parasuis, in this study, Glaesserella parasuis serovar 1, simply named HPS1”
  2. Line 72: The authors were mentioned about the primers, but I could not find in the table S1 file with this article. I found only the main text figures in the supplementary files. The authors must provide the primers details in the supplementary files.

Response: Something happened during submission. We reloaded Table S1 this time.

  1. Line 132: 11 stains contained the HPS_06257? Table 1 showed only 10 (+) symbols? Please verify it.

Response: Thank you, it has been verified in the revised Table 1.

  1. Line 184, 205, 212, 219 and 229: simplify as HPS5 than HPS_06257-expressing stains the results and discussion part.

Response: We appreciated the reviewer’s suggestion. To make this clear, we revised the results 3.3 and 3.4, in which HPS5 (an HPS_06257-expressing strain) and HPS11 (an HPS_06257-null strain) were used as representative strains. The changes have been highlight with yellow color.

  1. Figure 4 and 5 not in the chronological order, verify it.

Response: We generated Figure 4 by using antiserum from survival assay. Figure 5 was generated by using anti-HPS_06257 antibody that generated in result 3.2.

  1. Line 268: The authors must provide the Multiple sequence alignment and the percentage of identity in the supplementary section or need citation regarding the sentence "strong identity shared by the HPS_06257".

Response: We got this information by the BLASTP analysis. The results showed all of the hit sequences shared more than 99% identity. We had revised this sentence as “Regarding the strong identity shared by the protein sequences of different strains (based on the BLASTP analysis, data not shown),…”

  1. Discussion needs the implementation of the study and benefits of the animals welfare.

Response: We thank the reviewer to point this out. We have further defined this in the discussion as “In summary, this is the first study to demonstrate that antibody-dependent phagocytosis is involved in the protective effects of HPS_06257. These finding extend our understanding of how antibody-dependent phagocytosis may contribute to the immune protection afforded by other outer membrane proteins. Thus, future studies are needed to further verify this. In the meanwhile, they also provide insight into the protective antigens of G. parasuis and useful information for the development of a subunit vaccine to prevent G. parasuis infection.”

Minor corrections
1. Table 1: Write properly "Whole", some in small letters

Response: We had replaced it with Whole genome in the revised Table 1.

  1. Line 175-179 in page 6 and page 11: Error "HPS_0625" correct and verify it.

Response: Yes, you are correct. It has been improved.

Round 2

Reviewer 3 Report

The authors made few corrections and changes according to the reviewer comments, but still, the authors not responded (primer list Table S1) and not corrected (Table 1, supplementary files, "HPS_0625") in few places. So, I suggest the authors should to verify again all the given comments.

Table 1: PalA and Omp2 genes were missing in the supplementary information or original images provided by the authors.

Table 1: Write PalA and Omp2 genes in the table title/legend description as used as control or reference from the previous article.

As I requested the primer table for the revision, but again I could not find the Table S1 in the revised article. Authors should check the files in the submission system and provide the Table S1 primer lists.

Table 1: Revised supplementary files or original images had the same error file. 11 stains contained the HPS_06257? Table 1 showed only 10 (+) symbols? Please verify it.

Minor comments

Table 1: Write properly "Whole," HPS412_NZ_CP041334 small letters.

Figure 2 legend: Error "HPS_0625" change to "HPS_06257" correct and verify it. (revised article line numbers: 190, 191, 192, 193, 194).

Introduction: Provide HPS expansion in the introduction.

Corrections: The HPS_06257 was indicated in italics and few places are not. Correct it accordingly and change throughout the publications.
